# Heuristic-enabled active machine learning: A case study of predicting essential developmental stage and immune response genes in *Drosophila melanogaster*

**Olufemi Tony Aromolaran**[1,2]\*, **Itunu Isewon**[1,2], **Eunice Adedeji**[2,3], **Marcus Oswald**[4,5], **Ezekiel Adebiyi**[1,2], **Rainer Koenig**[4,5], **Jelili Oyelade**[1,2]\*

1 Department of Computer & Information Sciences, Covenant University, Ota, Ogun State, Nigeria, 2 Covenant University Bioinformatics Research (CUBRe), Covenant University, Ota, Ogun State, Nigeria, 3 Department of Biochemistry, Covenant University, Ota, Ogun State, Nigeria, 4 Integrated Research and Treatment Center, Center for Sepsis Control and Care (CSCC), Jena University Hospital, Am Klinikum, Jena, Germany, 5 Institute of Infectious Diseases and Infection Control, Jena University Hospital, Am Klinikum, Jena, Germany

\* ola.oyelade@covenantuniversity.edu.ng (JO); olufemi.aromolaran@stu.edu.cu.ng (OTA)

## Abstract

Computational prediction of absolute essential genes using machine learning has gained wide attention in recent years. However, essential genes are mostly conditional and not absolute. Experimental techniques provide a reliable approach of identifying conditionally essential genes; however, experimental methods are laborious, time and resource consuming, hence computational techniques have been used to complement the experimental methods. Computational techniques such as supervised machine learning, or flux balance analysis are grossly limited due to the unavailability of required data for training the model or simulating the conditions for gene essentiality. This study developed a heuristic-enabled active machine learning method based on a light gradient boosting model to predict essential immune response and embryonic developmental genes in *Drosophila melanogaster*. We proposed a new sampling selection technique and introduced a heuristic function which replaces the human component in traditional active learning models. The heuristic function dynamically selects the unlabelled samples to improve the performance of the classifier in the next iteration. Testing the proposed model with four benchmark datasets, the proposed model showed superior performance when compared to traditional active learning models (random sampling and uncertainty sampling). Applying the model to identify conditionally essential genes, four novel essential immune response genes and a list of 48 novel genes that are essential in embryonic developmental condition were identified. We performed functional enrichment analysis of the predicted genes to elucidate their biological processes and the result evidence our predictions. Immune response and embryonic development related processes were significantly enriched in the essential immune response and embryonic developmental genes, respectively. Finally, we propose the predicted essential genes for future experimental studies and use of the developed tool accessible at http://heal.covenantuniversity.edu.ng for conditional essentiality predictions.

**Data Availability Statement:** The data used for model evaluation is publicly available on UCI machine learning repository. The source code for

the heal application is available at https://github.com/phemmy2k2/conditional-essentiality. The data can be accessed through the link to the data repository as shown below: https://zenodo.org/record/8117236.

**Funding:** 1. Deutsche Forschungsgemeinschaft (https://www.dfg.de/) within the project KO 3678/5-1, and the German Federal Ministry of Education and Research (BMBF, Fkz 01EO1002, 01EO1502 and 13N15711) 2. World Bank awarded to Covenant Applied Informatics and Communication Africa Centre of Excellence (CApIC-ACE) through the ACE Impact Project (2019 – 2024) The funders had no role in study design, data collection and analysis, decision to publish, or preparation of the manuscript.

**Competing interests:** The authors have declared that no competing interests exist.

## Introduction

A gene is defined as absolute essential if its loss of function causes infertility or death in an organism or cell. There are a few computational approaches for predicting gene essentiality including homology search and evolutionary analysis-based approach [1], constraint-based methods [2], and machine learning (ML) approaches [3, 4]. Conditionally essential genes are genes that are essential in a particular condition but non-essential in another condition.

Conditional essentiality has predominantly been defined in terms of growth conditions [5, 6]. Recent systematic studies of gene essentiality revealed that two sets of essential genes exist; core essential genes that are always required for viability, and conditional essential genes that vary in essentiality in different genetic and environmental contexts [7]. The variability in essentiality depends on the phenotype being assessed (lethality, reproduction, growth and/or development), the species in which the gene is encoded and environmental/growth conditions [8, 9]. Costanzo and colleagues posited that environments often affect genes with a close functional relation to the pathways that are perturbed by a condition [10].

Another cause of variability in gene essentiality is experimental conditions such as temperature, pH, nutrient availability and/or, potentially, exposure to pathogens or microbes. Conditional essentiality has been linked to genetic factors. Some studies that systematically compared gene essentiality among closely related yeast isolates identified modifier loci that alter gene essentiality [11, 12]. Genetic factors also give rise to a phenomenon known as synthetic lethality where the loss of one of two genes that perform similar functions could render non-essential genes essential and essential genes dispensable [13]. More recently, it has become evident that gene essentiality also depends on the ability of cells to adaptively evolve and proliferate despite the inactivation of an essential gene, suggesting that essentiality is not a property of genes, but of cellular functions [13].

Curation of experimentally identified essential genes in developmental stages has been documented in some model organisms such as *Drosophila melanogaster* [14]. A set of essential genes are also required when a foreign body invades a host organism, this results in the immune response condition [15, 16]. Identifying a comprehensive set of essential genes in both developmental stage and immune response condition will be beneficial for identifying potential novel drug and insecticidal targets that overcomes the current drug and insecticide resistance in the fight against some diseases such as malaria [17].

Constraint-based methods such as Flux balance analysis have been used to identify conditionally essential genes [18]. However, the constraint-based methods have some drawbacks, which includes inability to identify non-metabolic genes and cannot be used to investigate genome-scale metabolic reactions under transient dynamic states without including data on enzyme kinetics [19, 20].

TnseqDiff is another commonly used computational approach for conditional essentiality prediction [21]. TnseqDiff utilizes two steps to estimate the conditional essentiality for each gene in the genome. First, it collects evidence of conditional essentiality for each insertion by comparing read counts of that insertion between conditions. Second, it combines insertion-level evidence to infer the essentiality for the corresponding gene. [21]. One of the major limitations of this approach is that transposon sequencing (Tn-seq) data is only available for a few model organisms thereby limiting the approach to bacteria. Owing to the genetic similarities and conserved pathways between *D. melanogaster* and mammals, the use of the Drosophila model as a platform to unveil novel mechanisms of infection and disease progression has been widely investigated [22] including host-pathogen interaction studies [23, 24].

Manimaran and others made the first attempt to explain the conditional essentiality of genes using the ML approach [25]. They obtained the protein interaction dataset from

predictions of genome-wide functional linkages in E. coli which contains 3,682 proteins and 78,048 interactions. Three centrality features (Degree, closeness, and betweenness) were used with an SVM model. The study focused on growth conditions determined based on the expression of the genes in a microarray dataset. They predicted 1192 bacteria genes to be conditionally essential across 61 growth conditions. An extensive experiment was conducted to obtain the essentiality status of the genes used to train the ML model which is a very expensive, challenging, and time-consuming approach.

Computational prediction of conditional essentiality research is an open problem that is gaining wide attraction in recent years. Recent reviews have identified prediction of conditionally essential genes as a major limitation of the ML approaches so far [19, 20]. This is a major limitation because there is no sufficient labelled data to train ML techniques for predicting conditional essentiality. This study is motivated by the challenge posed by data that could not be manually annotated by experts or require experiments for annotation as found in experimental studies, an example is identifying gene function from sequence information or predicting conditionally essential genes. Therefore, we sought to develop a ML technique that is capable of reliably predicting conditionally essential genes in both model and non-model organisms.

Active machine learning (AL) techniques have been used for annotating unlabelled data based on the limited data as seen in image recognition [26], activity recognition [27], and text labelling [28]. A traditional AL algorithm was presented by [29, 30]. Active learning algorithms are iterative sampling schemes, where a classification model is adapted regularly by feeding it with new labelled samples corresponding to the ones that are most beneficial for the improvement of the model performance. The new labelled samples used to improve the model performance are obtained using a sample selection strategy. A sample selection strategy describes the techniques used by the active learning procedure to select the most valuable points to be manually labelled. Some of the commonly used strategies are Uncertainty-based [31], Committee-based [32, 33] and Expected Impact [34] selection strategies.

Uncertainty-based AL technique is the most widely used. The challenge with the application of the existing AL techniques in bioinformatics is that most biological data require extensive literature search and experiments for annotation which is time-consuming and very expensive. Therefore, the use of AL techniques for annotating biological data when there is limited label data requires an innovative approach. This study replaces the human component of AL with a heuristic component to enable the application of AL to predict conditionally essential genes and pave a way for broad application of heuristic-enabled active learning to solve challenging problems in biomedical research. To our knowledge, this study is the first to apply machine learning method to conditional essentiality prediction.

## Methods

### Defining benchmark models and datasets

To benchmark the HEAL technique, the sampling query strategy used was replaced with a random selection technique which is hereafter referred to as the RandAL technique. In addition, the traditional AL technique that uses the uncertainty query strategy was implemented hereafter referred to as the UncAL technique. The RandAL technique trains the base classifier using the labelled data and the trained classifier was used to pre-label all the samples in the unlabelled set. Subsequently, a specified batch (n = 20) of pre-labelled samples were randomly selected and presented to the expert for manual correction of the pre-labelled samples. The manually corrected labelled samples are then added to the labelled dataset for the next iteration of the annotation until a stop criterion is reached. The stop criterion in this experiment is

**Table 1. Description of the real-world datasets for model validation.**

| Dataset | Number of Instances | Positive samples size | Negative samples size | Number of Attributes |
|---------|--------------------|-----------------------|-----------------------|----------------------|
| Breast cancer | 699 | 241 | 458 | 10 |
| Credit rating-A | 690 | 307 | 383 | 14 |
| Credit rating-G | 1000 | 700 | 300 | 24 |
| Diabetes | 768 | 268 | 500 | 8 |
| Heart disease | 270 | 150 | 120 | 13 |

when 90% of the entire dataset has been labelled. For the UncAL technique, the random selection of query samples was replaced by the uncertainty technique. The three techniques (HEAL, UncAL, and RandAL) were evaluated based on five real-world datasets retrieved from the UCI Machine Learning Repository [35]. The real-world datasets are diabetes, breast cancer, heart disease, and credit scoring (Australia and German) datasets. Table 1 presents the description of the real-world datasets. For each of the five datasets, 20 percent of the dataset was randomly selected as the labelled set and the class label was excluded from the remaining 80 percent which was designated as the unlabelled set.

## Datasets for conditional essentiality

This study sought to develop a technique for annotating unlabelled data when there is limited label data to train an ML model. Annotating genes that are essential in a given condition presents a problem of sparsity of labelled data [36]. For the purpose of evaluation of the developed technique, two categories of conditions in *D. melanogaster* were evaluated, these are the embryonic developmental condition and immune response condition. A total of 161 and 343 genes were collected from FlyBase as essential in embryonic developmental condition and immune response conditions respectively, while 12,058 and 11,993 genes were assigned as the negative class label samples in embryonic developmental condition and immune response conditions respectively. The essential embryonic genes were queried from Flybase using the term "lethal—all die during embryonic stage" which implies the organism died when the genes were mutated during the embryonic stage. For immune response labelling, Drosophila phenotypic data "allele_phenotypic_data_fb_2020_02.tsv" was downloaded from Flybase. Lethal and immune response were used to filter the phenotype of the genes. The most common technique used for identifying the physical trait of the genes was transposon mutagenesis. 21 genes were found to be essential in both conditions as shown in Fig 1B.

The training data for embryonic developmental condition comprised of 80 positive and 1600 negative samples which were randomly selected to represent the labelled data while the remaining samples represent the set of unlabelled data for the active learning analysis. 144 of the 343 essential immune response genes and 2880 of the non-essential immune genes were also randomly selected to represent the labelled data while the remaining samples represent the set of unlabelled data.

## Feature generation

Feature quality is a major factor in the development of a ML model for predicting essential genes. A total of 50,334 features were generated based on broad range of features derived from (1) gene sequence, (2) protein sequence, (3) functional domains of the proteins, (4) gene sets from Gene Ontology (GO), (5) the number of homologous sequences, (6) topology properties from protein-protein interaction networks, and (7) subcellular localization of the protein (Fig 1A). Protein and gene sequences were downloaded from Ensembl [37, 38] using BioMart [39].

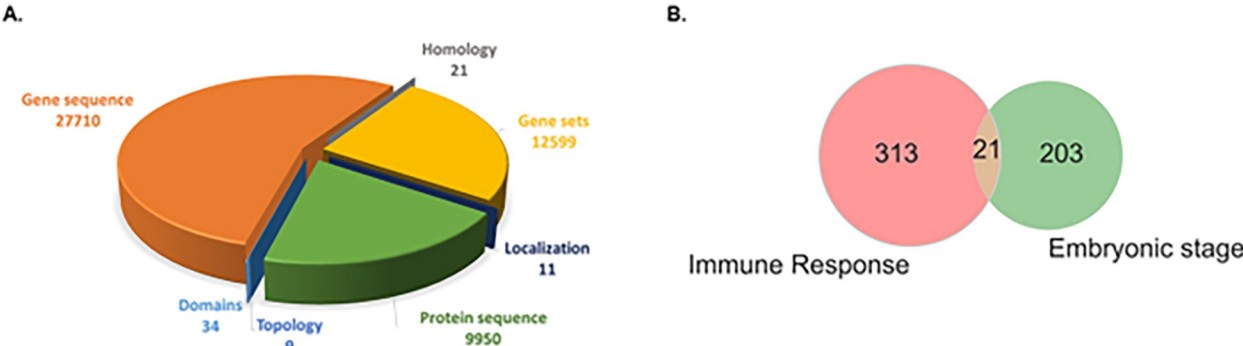

**Fig 1. Distribution of the features and class label for conditional essentiality prediction in *D. melanogaster*. A.** The generated features included intrinsic (e.g. protein and DNA sequence) and extrinsic features (e.g. topology of co-expression and protein-protein interaction networks). The number of features derived from individual categories is shown below the various categories. **B.** Venn diagram shows the total number of essential genes in each condition obtained from FlyBase and the number of genes essential in both conditions.

For deriving the protein and gene sequence features (features in categories 1 and 2), various numerical representations characterizing the nucleotide and amino acid sequences and compositions of the query genes were calculated using seqinR [40], protr [41], CodonW [42] and rDNAse [43]. Using seqinR [40] the number and fraction of individual amino acids and other protein sequence features including the number of residues, the percentage of physico-chemical classes and the theoretical isoelectric point were calculated. Further protein sequence features were obtained using protr [41] including autocorrelation, Conjoint Triad Descriptors (CTD), quasi-sequence order and pseudo amino acid composition. CodonW [42] was used to calculate gene characteristics like gene length and GC content but also frequencies of optimal codons (frequency of codons favored by natural selection, see [44]) and the effective number of codons. Using rDNAse [43] gene descriptors like auto covariance or pseudo nucleotide composition, and *kmer* frequencies (n = 2–7) were calculated.

For deriving domain features (feature category 3), BioMart was used to obtain protein family (*pfam*) domains, number of coiled coils, the prediction of membrane helices, post-translational modifications, β-turns, cofactor binding, acetylation and glycosylation sites, trans membrane helices and signal peptides. In addition, the number and lengths of UTRs were obtained using BioMart. For features obtained from gene sets defined by Gene Ontology (feature category 4), gene sets of all GO terms including biological process, cellular localization and molecular function were obtained from Ensembl (version 102, released in Nov, 2020) [37, 38]. Gene sets were removed if they showed high redundancy according to the following method. The gene overlap of each pair of gene sets A and B was quantified by Jaccard similarity coefficients,

$$J(A, B) = \frac{|A \cap B|}{|A \cup B|} \quad (1)$$

Pairs with J(A, B) above a threshold (threshold = 0.3) were included in the model and represented as an undirected graph, G = (X, E), with the gene sets as vertices X and the pairs above the threshold as edges E. A linear model was set up with a constraint to select at most one of the vertices of an edge:

$$Xi + Xj \leq 1, \qquad for\ every\ \{i, j\} \in E \quad (2)$$

$$Xi = 0,\ or\ Xi = 1, \quad for\ 1 \leq i \leq n \quad (3)$$

with the objective function to *maximize*

$$\sum w_i X_i$$

where, $w_i$ is the weight of a gene set. The weight is derived from its significance (p-value) and calculated as $1 - log10(p\text{-}value)/100$. This maximization was done employing linear integer programming solved using Gurobi (version 7.5.1, https://www.gurobi.com). With this, we formulated the optimization problem to select at most one gene set from each pair in such a way that the overall number of non-redundant gene sets was maximized. This optimization problem was formulated as a mixed integer linear programming problem and solved using Gurobi (version 7.5.1, https://www.gurobi.com). A gene list was generated for each query gene according to a protein association network obtained from the STRING database [45]. The gene list for a gene is the set of all adjacent genes in the protein association network. A gene set enrichment test was performed employing Fisher's exact test and the negative *log10* of the *p*-value was used as a feature.

The number of homologous proteins (feature category 5) was obtained by blasting the protein sequence of the query protein against the complete RefSeq database [46] using PSI-BLAST [47]. The number of proteins found with e-value cutoffs from 1e–5 to 1e-100 were used as features. Topology features (feature category 6) were computed using the NetworkX [48] library in Python. Protein association data was downloaded from STRING [45] and an undirected network was constructed. From this, degree, degree distribution, closeness centrality, eigenvalue centrality, betweenness centrality, harmonic centrality, subgraph centrality, load centrality and Page rank as topological features were computed for each gene. To note, the harmonic centrality of a node *g* is the sum of the reciprocal of the shortest path distances from all other nodes to *g*. The higher the value, the higher the centrality [49]. The subcellular localization of proteins (feature category 7) was derived using DeepLoc [50]. DeepLoc predicts the likely location of a protein within a cell by assigning probability scores to eleven eukaryotic cell compartments (cytoplasm, nucleus, extracellular, mitochondria, plasma membrane, ER, chloroplast, Golgi apparatus, lysosome, vacuole and peroxisome). In total we generated 50,334 features.

## Data normalization and feature selection

The dataset for conditional essentiality prediction consists of thousands of features from different categories with different range of values. Therefore, the data requires to be normalized and prepared for ML. All the features were merged into a single table followed by a *z*-score transformation of each feature to normalize the data. In addition, redundant highly correlating features with Pearson correlation coefficients > 0.70 were removed to avoid multicollinearity which introduces a bias in the analysis and extrapolation is likely to be seriously erroneous [51, 52]. If more than two features are highly correlated, then the one with the highest correlation with the target class was selected.

To overcome the class imbalance problem when training the classifier, Synthetic Minority Over-Sampling Technique (SMOTE) was used. SMOTE is a technique that creates synthetic, non-duplicated samples of the minority class, thereby making the total samples in both minority and majority classes to be equal [53]. For each minority class observation, SMOTE calculates the *k* nearest neighbours and randomly creates multiple synthetic samples between the observation and the nearest neighbours depending on the number of oversampling needed.

For each iteration and based on the labelled set, we performed two steps for feature selection prior to training of the machines. First, we applied an embedded approach based on Random Forests as suggested by [54] for feature selection. Each tree in the forest was initialized by bootstrapping from the training data to train a baseline model. Its performance was estimated

using the out-of-bag (OOB) samples from the training data. Then, the values of one feature were randomly shuffled, keeping all other features the same, yielding permutated data. The permutated dataset was applied to the learned model and its performance was evaluated. Finally, the difference between the benchmark score from the baseline model and the score from the permutated model was calculated to determine the importance of the feature [55]. By this, we ranked all features and selected the top 400 features for training the downstream classifier.

## Heuristic-enabled active machine learning

In this study, Light GBM, an ensemble model was used as the classifier for the active learner due to its high prediction accuracy and fast execution time [55]. Also, in recent studies, ensemble models such as Random forest and Extreme gradient boosting have shown to have a good performance on numerical data from biological sources [3, 56]. Due to the small size of the labelled data, 5-fold CV was used during the training of the classifiers. The hyper-parameter settings for the classifier was set according to the optimal settings obtained in our previous study [56] where *n_estimators = 600, learning_rate = 0.05, num_leaves = 32, colsample_by-tree = 0.2, reg_alpha = 3, reg_lambda = 1, min_split_gain = 0.01 and min_child_weight = 40*.

The traditional AL algorithm was modified by replacing the human component with a heuristic function that uses a threshold specified by the user to filter queried samples. The sampling query function was also modified to use the *certainty* sampling technique proposed by this study instead of the widely used uncertainty technique. The certainty technique is the reverse of the uncertainty method. Unlike the uncertainty method that selects samples close to the classification or decision boundary as queries for the human expert, the certainty technique selects samples with high prediction confidence, these are samples with high prediction probability for the positive class and very low prediction probability for the negative class. Typically, the prediction probability is between 0 and 1 and by default, ML algorithms set their classification boundary as 0.5. It classifies all samples with a prediction probability below 0.5 as negative samples while those with a level of 0.5 and above are classified as positive samples. However, the classification boundary was set to 0.6 for this analysis when it was observed that the data was biased towards the positive samples and resulting in a high false positive rate.

The sample selection strategy described in the heuristic function introduced by this study is based on a cut-off that is dynamically assigned at each iteration according to the distribution of the classes in the pre-labelled dataset. The distribution of the prediction probability of the positive (0.6–1.0) and negative (0.0–0.59) samples as obtained from the automatic annotation is represented as quantiles. Samples in the first quartile for the negative distribution ($Q^{-1}$, closer to 0) and samples within the fourth quartile of the positive distribution ($Q^{+4}$, closer to 1) were selected by the heuristic function for further filtering. The heuristic function contains a threshold set by the user to exclude samples below the threshold and append the samples with values above the threshold to the labelled data. This further increases the classifier's prediction power which has a similar impact as selecting samples closer to the classification boundary, requiring humans to refine the automatic annotation by the classifier manually. A threshold of 0.9 was chosen by this study which ensures samples very close to the positive samples in the labelled data are annotated as positive. The complete implementation is described in Algorithm 1 and 2 and the schematic workflow of the Heuristic-Enabled Active Learning (**HEAL**) is shown in Fig 2.

```
Algorithm 1: Heuristic-Enabled Active Learning Algorithm
Inputs
```
- Initial training set $X^\alpha = \{x_i, y_i\}_{i=1}^{l}$ (X∈χ, α =1)
- Pool of candidates $U^\alpha = \{x_i\}_{i=l+1}^{l+u}$ (U∈χ, α =1)

```
-  Threshold = confidence cut-off specified by the user.
1: repeat
2:  Train a model with the current training set X^α.
3:  for each candidate in U^α do
4:    Score = Evaluate a user-defined classification model
5:  end for
6: P^α = U^α + Score
7:  Rank the candidates in P^α according to the score of the classifica-
tion model
8: S^α = Heuristic component (P^α, threshold)
9:  Add the batch to the training set X^{α+1} = X^α∪S^α
10:  Remove the batch from the pool of candidates U^{α+1} = U^α\S^α
11: α = α+1
12: until a stopping criterion is met.
Algorithm 2: Heuristic component (P^α, threshold):
1:  Divide P^α into P^+ and P^- sets according to ranked score
2:  Compute quartiles for P^+ and P^-: Q_i = i/4(n+1)^{th} term|(1 ≤ i ≤ 4, n = |U|)
3:  Dynamically estimate the filtering cut-off for P^+ and P^-
4:   if min (P^{+Q4}) > threshold
5:   sup = min (P^{+Q4})
6:   else:
7:   sup = threshold
8:  Set cut-off for P^-: inf = max (P^{-Q1})
9:  Filter P^+ and P^- based on sup and inf respectively where q = the
batch size of selected points S.
10: Return filtered sample points S^α = {x_k, y_k}_{k=1}^q to the main function
```

## Gene set enrichment analyses of the predicted essential immune response genes

To discover the biological and functional knowledge of the genes predicted to be essential for embryo developmental stage and immune response conditions in *D. melanogaster*, gene set enrichment analysis was performed using g:Profiler based on the Ensembl database version 102 [57]. The SCS algorithm with default settings as described by [57] was used to correct for multiple testing and the significance threshold was set to $p = 0.05$. The term size of the selected enriched gene sets was set between 3 and 500 to filter out too specific and too general gene sets.

# Results

## Evaluation of HEAL and benchmark models

A main hypothesis of this study was that a comparable performance could be achieved by replacing the human component in the AL model with a heuristic function to eliminate the high cost and time involved in using the traditional AL model. To evaluate the performance of HEAL to existing traditional AL models, we implemented Uncertainty AL, which is the traditional AL model based on uncertainty sample selection method with human component and Random AL, which is also a traditional AL model based on random sample selection method. Five publicly available datasets were applied to the three AL models. The results show that HEAL performed comparatively better than UncAL and has superior performance when compared with RandAL (Fig 3). HEAL has a significantly lower running time compared to both UncAL and RandAL across the five datasets (Fig 3D). The reason for the low running time is because HEAL does not require an expert for its annotation which is associated with

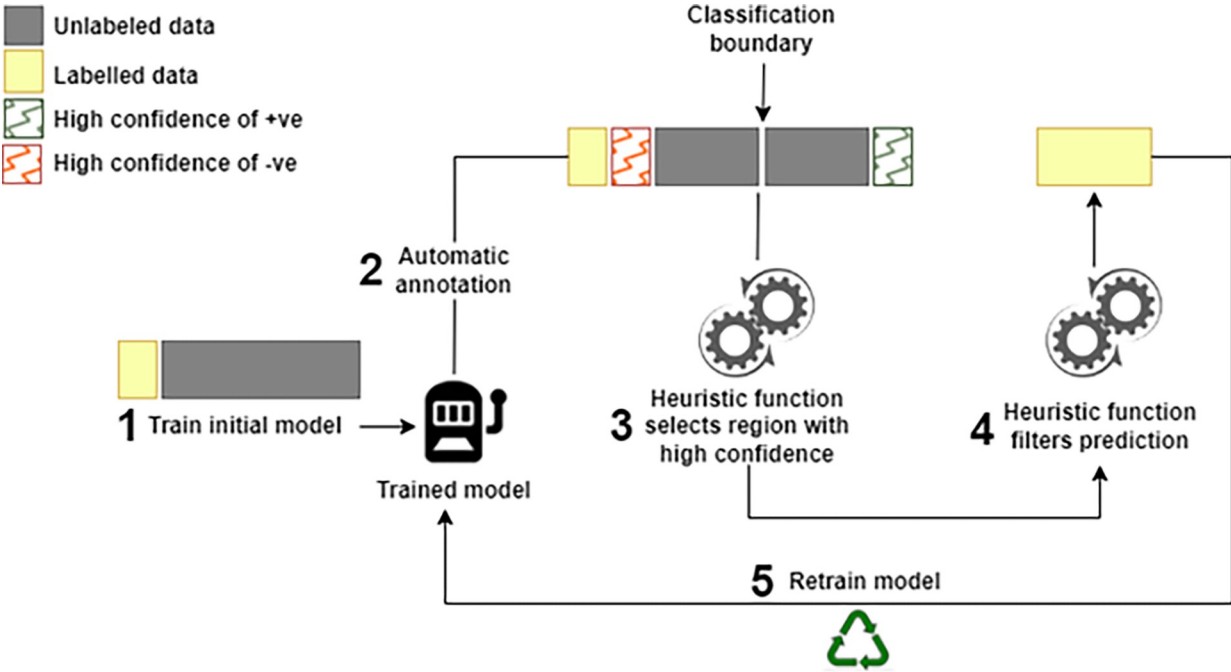

**Fig 2. Heuristic-enabled active learning implemented to label conditionally essential genes.**

computational delay prior to each iteration in the process. A stratified view of the predictions from the three techniques across the five datasets based on the confusion matrix is presented in S1 Fig.

### Prediction of conditionally essential genes in *Drosophila melanogaster*

To evaluate the proposed model for conditional essentiality prediction, two conditions based on immune response and developmental stage conditions were examined. For the developmental stage conditions, a total of 53 genes were predicted as essential in the embryonic stage of *D. melanogaster* after five iterations with five of the predicted genes annotated as such in FlyBase. We performed gene set enrichment analysis to elucidate the biological processes enriched in the predicted genes. We found several growth and morphogenesis processes, such as post-embryonic animal morphogenesis and post-embryonic animal organ development (Fig 4A) indicating the need for these specific growth processes for the organism to develop from the embryonic stage into the larva stage. Table 2 shows the list of top 10 genes predicted to play essential roles during the embryonic stage of *D. melanogaster* and the complete list of predicted genes is shown in S1 Table.

For the immune response conditions, a total of 10 genes were predicted as essential for these conditions in *D. melanogaster* after eight iterations with 6 of the predicted genes annotated as such in FlyBase. Strikingly, the enrichment analysis of the predicted immune response genes revealed that immune and defense response related processes are significantly enriched in the predicted genes (Fig 4B) which implies that the four novel immune response genes would be good candidates for further experimental validation. Table 3 shows the list of the 10 genes predicted to play essential roles during the embryonic stage of *D. melanogaster*. Genes highlighted in red were found to be annotated as essential immune response genes in FlyBase.

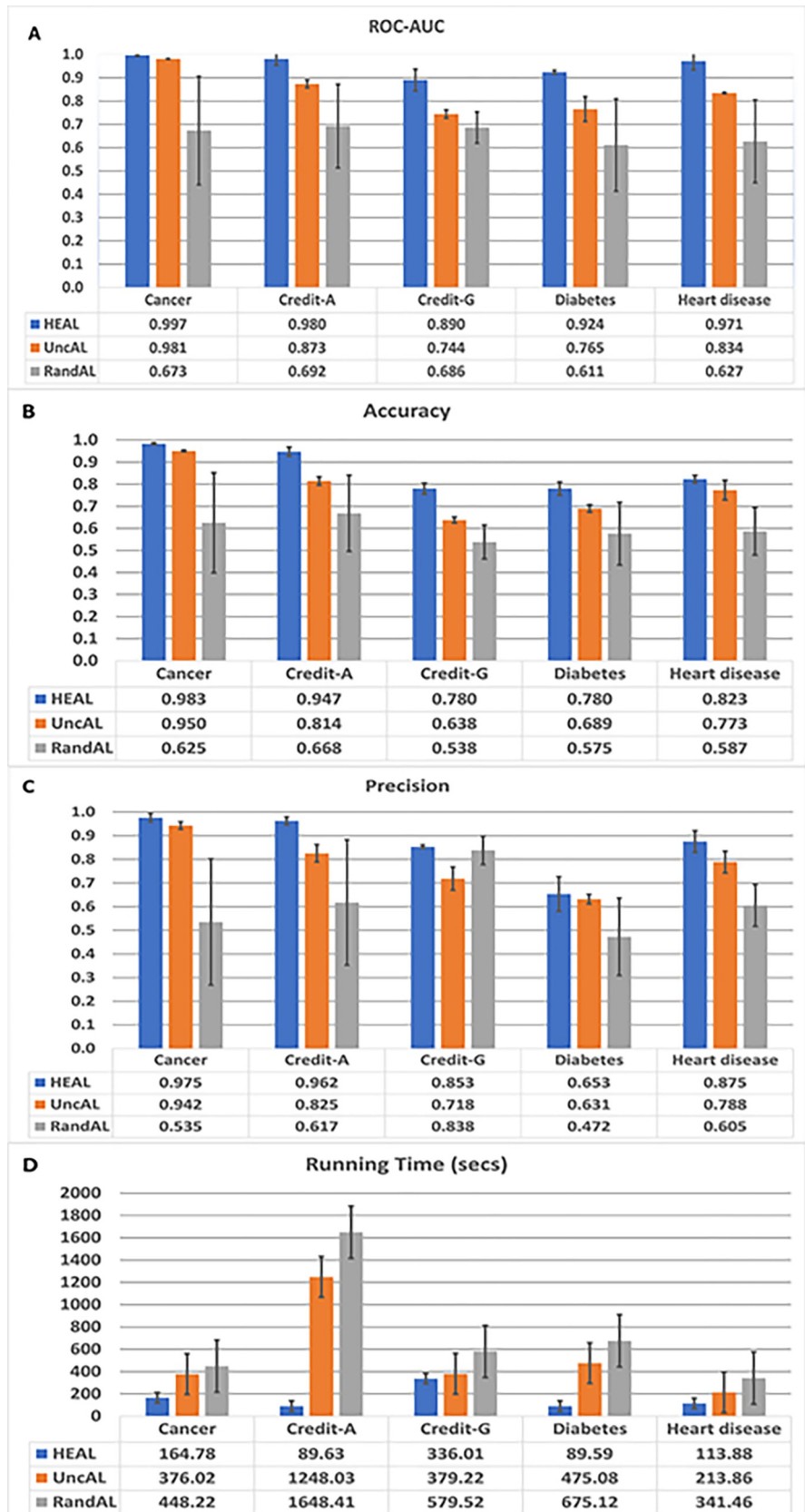

**Fig 3. Comparative analysis of HEAL and other techniques on five real-world datasets.** Results from HEAL technique show superior performance in terms: **A.** ROC-AUC **B.** Accuracy and **C.** Precision. **D.** HEAL has the lowest running time compared to other methods.

## Implementation of HEAL technique as a web resource

Lack of labelled data has been a challenge in bioinformatics research that has persisted for decades. Prediction of conditionally essential genes using a machine learning approach is an example of the numerous bioinformatics problems that remained intractable. Notably, this study developed the HEAL technique into a web application that provides a tool for bioinformatics analyses to annotate biological and non-biological data when there is limited labelled data for training a machine learning model. Django framework was used for the development of the web version of the HEAL technique. Django is a high-level Python framework that encourages rapid development and clean, pragmatic design. The framework integrates Hypertext mark-up language (HTML), Cascading stylesheet (CSS), and JavaScript seamlessly with Python. This application can be found on heal.covenantuniversity.edu.ng. Fig 5 presents the HEAL web application portal. The portal provides fields for uploading the labelled and unlabelled data in CSV format. After uploading the input data, the user has the option to first preview the statistical information about the input data or perform data annotation directly. The data statistics returned include the number of features, the ratio of labelled to unlabelled, and the ratio of the class label of the initial training data. The data statistics are displayed on the right-hand side of the screen. If an error occurs during the prediction, a log will appear specifying the error. The server was configured to reject the processing of files with sizes more than 4Mb to avoid overloading the server with large data. This implies that feature selection should be performed before using this tool for data with a large feature set.

A dropdown element provides a list of threshold values for the active learning component for selection by the user. The threshold determines the stringency level of the active learning

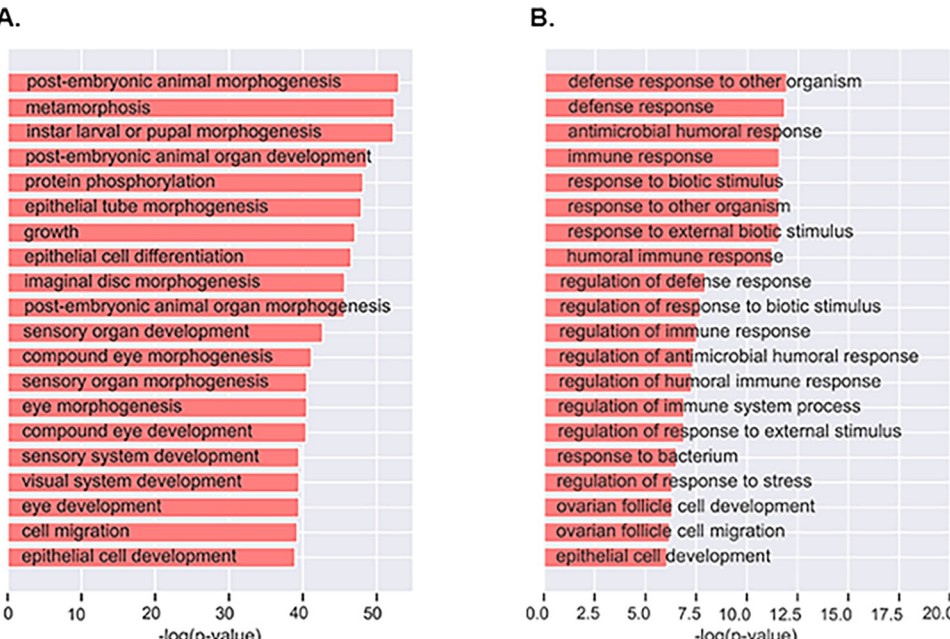

**Fig 4. Functional enrichment analysis of genes predicted by the HEAL model.** A. Biological process enriched in the predicted essential embryonic genes. B. Biological process enriched in the predicted essential immune response genes.

**Table 2. Top 10 predicted essential embryonic stage genes.**

| FlyBase ID | Gene Name | Gene Description |
|---|---|---|
| FBgn0265623 | Su(z)2 | Suppressor of zeste 2 |
| FBgn0283427 | FASN1 | Fatty acid synthase 1 |
| FBgn0287184 | FASN3 | Fatty acid synthase 3 |
| FBgn0286784 | TER94 | Transitional endoplasmic reticulum 94 |
| FBgn0003315 | satDNA | satellite DNA, unknown function |
| FBgn0286785 | scb | scab |
| FBgn0286786 | hoip | hoi-polloi |
| FBgn0036448 | mop | myopic |
| FBgn0265434 | zip | zipper |
| FBgn0036980 | RhoBTB | Rho-related BTB domain containing |

component with the default threshold value set to 0.9. After the server successfully finishes the processing, the output from the analysis which contains the dynamic cut-off for selecting high confidence samples, current classifier performance scores, and the number of samples added to the labelled set for each iteration are displayed in the result page shown in Fig 6. The base classifier performance indicates the confidence of the prediction. It is recommended to have a minimum of 90% base model accuracy for a reliable prediction.

## Discussion

As at the time of conducting this research, there are no studies found from literature reviewed that have successfully applied machine learning techniques to predict conditionally essential genes responsible for any condition. A related study used a semi-supervised ML approach to predict HIV dependency factors in humans using only network-based features from protein interaction databases. They reported a precision score of 85% at 60% recall [58]. Some of the top-ranked genes predicted as essential in immune response conditions are discussed below along with their functions with respect to their importance to the organism's immune response conditions.

The state-of-the-art AL techniques have consistently used the uncertainty method for sampling selection and present the queried samples to the expert for manual correction of the pre-labelled samples. For the next iteration, the manually corrected samples were added to the

**Table 3. Predicted essential immune response genes.**

| FlyBase ID | Gene Name | Gene Description |
|---|---|---|
| **FBgn0035976** | PGRP-LC | Peptidoglycan recognition protein LC |
| **FBgn0016917** | Stat92E | Signal-transducer and activator of transcription protein at 92E |
| **FBgn0041184** | Socs36E | Suppressor of cytokine signaling at 36E |
| **FBgn0043903** | dome | domeless |
| **FBgn0000250** | cact | cactus |
| **FBgn0034476** | Toll-7 | Toll-7 |
| FBgn0004364 | 18w | 18 wheeler |
| FBgn0002930 | nec | necrotic |
| FBgn0086358 | Tab2 | TAK1-associated binding protein 2 |
| FBgn0000229 | bsk | basket |

# Genes in bold typeface are annotated as immune genes in FlyBase

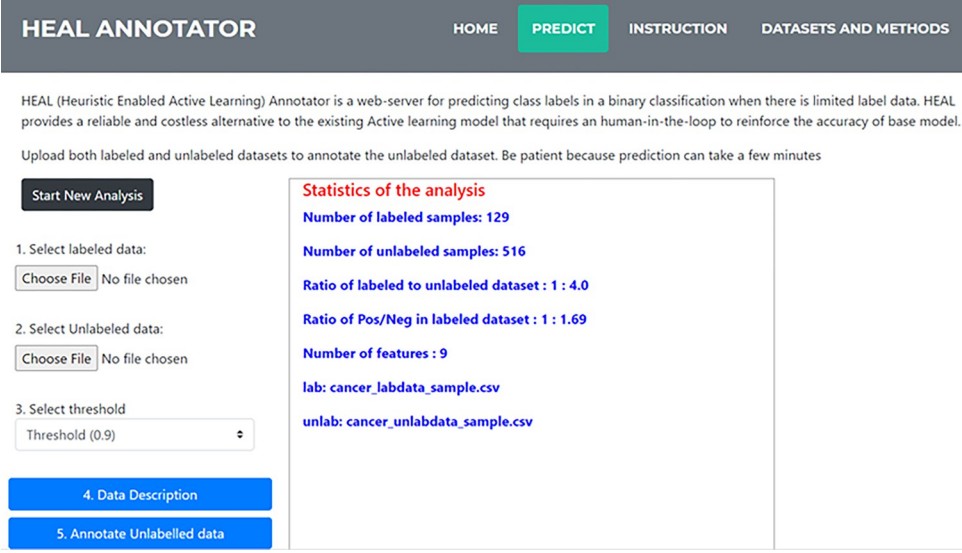

**Fig 5. Data description page of the HEAL annotator web application.**

labelled set and removed from the unlabelled set. The iteration was repeated until all samples in the unlabelled set have been completely labelled. In this study, heuristic-enabled active learning (HEAL) model, which replaces the human component of the traditional AL with a heuristic function, was developed. To benchmark the HEAL technique, the state-of-the-art AL

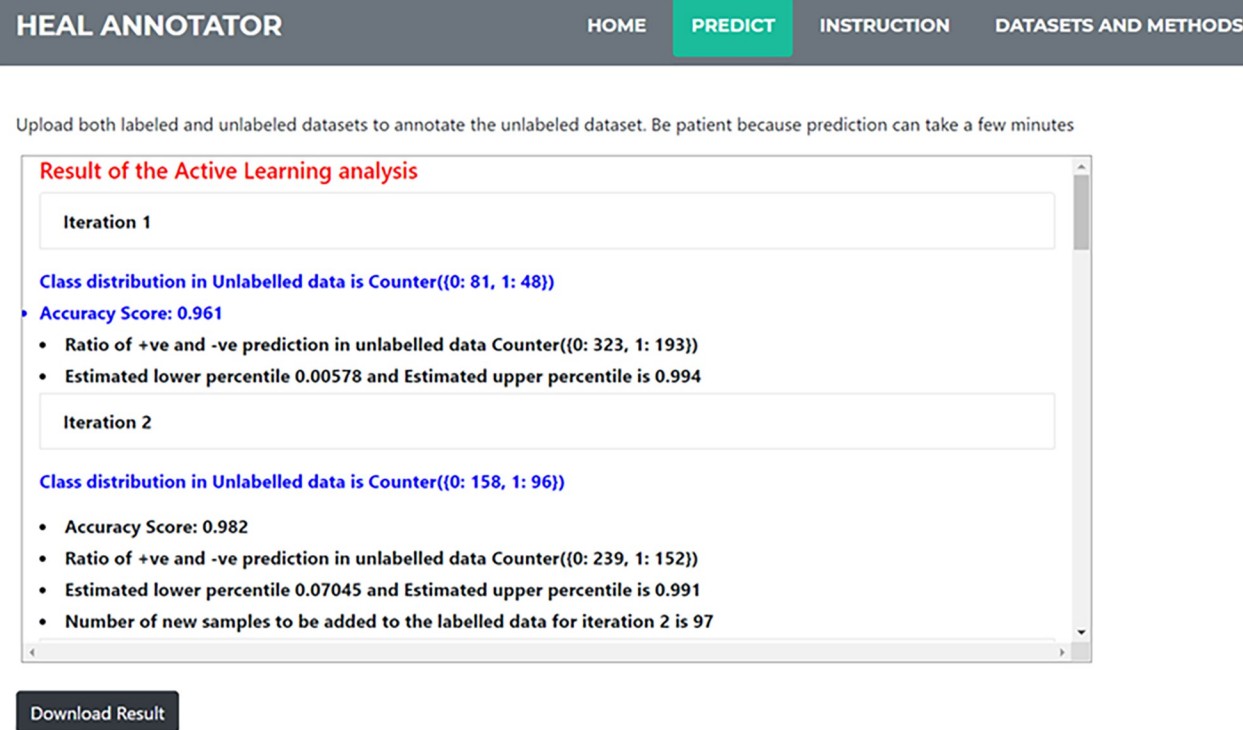

**Fig 6. Result page of the HEAL annotator.** User uploads their input files, and the program annotates the unlabelled dataset and presents the result to the user.

technique that uses the uncertainty method (UncAL) and random sampling method (RandAL) were implemented. Five publicly available datasets were applied to the three techniques and the HEAL technique performs better when compared to the UncAL technique. The RandAL technique outputted the least performance which means that informed selection of samples from the unlabelled set to be added to the labelled set is critical for the good performance of the AL techniques. The *certainty* technique introduced by this study also seeks to increase the prediction power of the ML model by selecting samples with high confidence based on the pre-labelling by the base classifier to be added to the labelled set. The ambiguity at the decision boundary of the model is gradually resolved as the prediction power of the ML model increases. This accounts for the good performance recorded by HEAL technique.

In comparing the running time for the evaluated techniques, the HEAL technique showed a significantly reduced running time. The low running time recorded by HEAL is a result of the replacement of the human component from the AL process with a heuristic function. The human expert is required to manually go over all the selected pre-labelled samples and correct them one after the other which will be cumbersome if the size of the unlabelled set is large. Replacement of the human expert with the heuristic function provided by the HEAL technique also eliminates the financial cost associated with employing an expert for manual annotation thereby making it a preferred choice for AL techniques in future studies. During the development of HEAL, only binary classification was considered which implies that HEAL cannot be directly applied to a multiclass classification problem. HEAL performed well on mixed (Categorical, Boolean, and Continuous) data types.

The choice of training samples for conditional essentiality prediction is a potential limitation of this computational approach. The chances of getting sufficient positive and negative samples of a specific condition to train an active learning model are very low because most experimental studies focus on identifying "what genes do" and not "what they did not do" and the function of several genes are yet to be completely known. The use of wrong training data will affect the accuracy of the active learning model. When validated using Flybase, the proposed HEAL model performs significantly well in immune response condition compared to the embryonic developmental stage condition. The choice of features used to build the model determines how well the model performs in varying conditions.

In the following, we discuss some of the top-ranked genes predicted as essential for embryonic stage and immune response conditions in *D. melanogaster* along with their functions based on findings from our literature study. Ten proteins were predicted by HEAL as important for immune response. Six of these genes: Peptidoglycan recognition protein LC (*PGRP-LC*, FBgn0035976); Signal-transducer and activator of transcription protein at 92E (*Stat92E*, FBgn0016917); Suppressor of cytokine signaling at 36E (*Socs36E*, FBgn0041184); Domeless (*dome*, FBgn0043903); cactus (*cact*, FBgn0000250); Toll-7 (*Toll-7*, FBgn0034476) are already annotated as immune response genes in the literature [59–63]. For example, Peptidoglycan recognition protein LC (*PGRP-LC*, FBgn0035976) encodes a transmembrane receptor that is recognized and bounded to diaminopimelic acid (DAP)- containing peptidoglycan [64]. DAP- containing peptidoglycan is a cell wall component found on Gram-negative bacteria and certain Gram-positive bacteria. Its binding to *PGRP-LC* during bacterial infection activates the immune deficiency signalling pathway [65]. This leads to the induction of antibacterial genes and phagocytosis [60, 66]. Mutations in *PGRP-LC* leading to a loss of function increases susceptibility to gram-negative bacterial infection [67]. Knockdown of *PGRP-LC* also increased the copy number of sigma virus and reduced the survival rate of Drosophila infected with sigma virus when treated with $CO_2$ [68], very likely to be linked to the cellular immune response to sigma virus infection. *Stat92E* is important in sustaining an effective balance between immune responses and also in inhibiting transcription of diverse immune

effector genes activated by Relish. *Stat92E* mutant flies have been reported to have higher bacterial clearance activities compared to the wild type. However, these mutant flies die upon bacterial infection [69]. This reveals the importance of *Stat92E* in regulating a balanced immune response.

The four other genes predicted as immune response genes were 18 wheeler (*18w*, FBgn0004364 or FBgn0287775); necrotic (*nec*, FBgn0002930); TAK1-associated binding protein 2 (*Tab2*, FBgn0086358), and basket (*bsk*, FBgn00002290). The *18w* encodes a member of the Toll-like receptor family involved in antibacterial humoral response. *18w* mutant flies (larvae) have been reported to have reduced expression of antimicrobial peptide genes and suffered increased lethality upon bacterial challenge [70]. However, in adult flies, *18w* mutant flies had expressed antimicrobial peptide genes at levels similar to the wild type [71]. This suggests that the role of *18w* in immune response is age or developmental stage-specific. [72] reported increased expression levels of *18w* in 4-week-old flies infected with *E. coli* compared to their 1-week-old infected counterparts. Also, the transcript level of *18w* was significantly correlated (r = 0.80) with the ability of the flies to clear the bacteria compared to their 1-week-old counterparts. This further emphasizes the age-specific importance of *18w* in immune response to bacterial challenge.

Similarly, *nec*, encodes a hemolymphatic serine protease inhibitor (*serpin*, *spn*)—*Spn43Ac* that negatively regulates the Toll immune signalling pathway [73, 74]. The *nec* mutant flies constitutively express *Drosomycin*, in response to fungal infection [74]. *Tab2* participates in the activation of the immune deficiency (*Imd*) signalling pathway through its interaction with the product of transforming growth factor (*TGF*) beta-activated kinase 1 (*Tak1*) [75]. dsRNA silencing of *Tab2* has been noted to block expression of an antibacterial peptide produced by *Imd* activation, and *JNK* activation by peptidoglycans [76]. Likewise, *Tab2* RNAi eliminated the induction of a broad range of immune response genes in S2 Drosophila cells [77]. Further to this, *bsk* encodes a serine/threonine-protein kinase, a key component of the *JNK* signalling pathway. Drosophila *bsk* RNAi knockdown mutants have been reported to completely lack clot melanization [78]. Also, *bsk* mutant Drosophila larvae failed to melanise eggs from the parasitoid *Leptopilina boulardi* [79]. These studies showed *bsk* as an important mediator for cellular immune response through melanisation.

HEAL predicted 53 genes as essential for development. The top ten genes include Suppressor of zeste 2 (*Su(z)2*, FBgn0265623); Fatty acid synthase 1 (*Fasn1*, FBgn0283427); Fatty acid synthase 3 (*Fasn3*, FBgn0287184); Transitional elements of the endoplasmic reticulum 94 kDa (*Ter94*, FBgn0286784); Scab (*scb*, FBgn0286785); Hoi-polloi (*hoip*, FBgn0286786); myopic (*mop*, FBgn0036448); Zipper (*zip*, FBgn0265434 or FBgn0287873); Rho-related BTB domain containing (*RhoBTB*, FBgn0036980); Shibire (*shi*, FBgn0003392). These are discussed as follows:

*Su(z)2* encodes a protein that is a functionally redundant homologue of the Polycomb Group (*PcG*) gene Posterior sex combs (*Psc*) protein [80]. *PcG* proteins are epigenetic regulators crucial in maintaining cell fate and stem cell function [81]. *Psc/Su(z)2* alongside Polyhomeotic (*PH*), Polycomb (*PC*), and *dRING* make up the Polycomb repressor complex 1 (*PRC1*) which play a role in ubiquitination of *H2A* [82]. *Su(z)2* restricts the proliferation and maintains the identity of the Cyst Stem Cell (*CySC*) in testis samples of Drosophila. It is also important for germline stem cell (GSC) maintenance and germ cell development, observed to act as a tumor suppressor [83]. *Su(z)2* disrupts dmyc auto-repression, Hence, it provides and maintains *Myc* levels required for embryonic growth and proliferation [84]. Similarly, [85] reported that only 17.4% of embryos from *Su(z)2* mutant flies emerged as adults compared to 91.5% adult emergence observed in wild type. These studies reveal the importance of *Su(Z)2* in the development of Drosophila from embryo to adults.

*Fasn1* and *Fasn3* encode fatty acid synthase involved in the biosynthesis of saturated fatty acids [86]. [87] reported that *Fasn–/–*mutant mice embryos died before implantation and the *Fasn+/–*embryos died at various stages of their development, hence, the importance of *fasn* in embryonic development. In Drosophila, *Fasn1* levels have been observed to steadily increase during embryogenesis (peaks at 13.5–18 h), and then decline at the end of the embryonic life [88]. While *Fasn1* is present in all larvae tissues, *Fasn3* is expressed in the cuticle, epidermis, muscle, and oenocytes of larvae [89]. [90] noted that Drosophila with RNAi targeting *Fasn3* in their oenocytes, produced embryo that did not mature into adults. Lethality in the offspring was observed either at the second/third larval transition stage 4–5 days after egg deposition, at the third larval stage or at the pupa stage. However, flies with RNAi targeting *Fasn1* in oenocytes produced viable offspring. This might be due to an incomplete RNAi effect, although *Fasn3* is oenocyte-specific in adult flies but *Fasn1* is not [91]. These studies reveal the importance of fatty acid synthase 1 in the development of Drosophila.

*Ter94* is a regulator of the ubiquitin proteasome system [92]. It is expressed in the embryo, in pupae, and in imago, but suppressed in the larvae stage of Drosophila [93]. Overexpression of *Ter94* RNAi in Drosophila third instar wing imaginal discs has been observed to cause pupal lethality [94]. Similarly, *Ter94* is important for oogenesis. [95] reported that embryos laid by female flies with germ-line clones of weak loss-of-function alleles of *Ter94* have reduced hatchability compared to the wild type. In turn, female flies with germ-line clones of a strong loss-of-function allele of *Ter94*, do not produce egg chambers [96]. *Ter94* regulates Bone morphogenetic proteins (BMPs) signalling during embryogenesis [97]. It also positively regulates Notch signalling [98]. Also, maternal knockdown of Ter94 caused significant 86% arrest in early stage 2 embryogenesis [99]. Hence, it is important for developmental events in the fly.

*Scb* encodes the α-PS3 Integrin. It regulates cell adhesion, signalling, polarity, and migration [100]. It is required for heart lumen formation [101]. S*cb* mutant flies have abnormal salivary glands, mislocalized pericardial cells and interrupted trachea [102]. It regulates pupal wing vein formation [103]. Also, mutations in *scb* reportedly resulted in impaired phagocytosis of apoptotic cells in Drosophila embryos [104]. These studies allude to the importance of *scb* during the development of Drosophila.

*Hoip* in Drosophila encodes a highly conserved RNA-binding protein [105]. *Hoip* mutant embryos have been reported to have aberrant myogenesis preventing them from emerging from the chorion after embryogenesis [106]. Hence, *hoip* is necessary for the initiation and maintenance of muscle structural gene expression during embryogenesis. Deficiency of *hoip* in mice has also been noted to cause embryonic lethality [107]. These studies portray *hoip* to be important during development in flies.

In turn, mop encodes a His domain protein-tyrosine phosphatase [108]. Depletion of mop impairs border cell cluster integrity and cell adhesion during oogenesis in Drosophila [109]. In Drosophila, *zip* encodes the non-muscle myosin II heavy chain. Zip mutant embryos have abnormal cell shape changes in the epidermis and incomplete dorsal closure [110]. Dorsal close in Drosophila embryo involves reorganization and contractions of the actin-myosin cytoskeleton within epithelial cells, thereby leading to the shaping of the embryo [111]. [112] reported that zip RNAi embryos had aberrant elongation (about 80% of zip RNAi embryo had < 50% of egg length, compared to the wild type which all had ≥70% egg length). Similarly, none of the zip RNAi embryo hatched 1 day after compared to the wild type in which >80% of the embryos hatched. The study revealed that the absence of zip leads to embryonic lethality. Hence, it is essential for embryo development.

*RhoBTB* is an atypical Rho GTPase. It is important for dendritic development in Drosophila with its knockdown in dendritic arborization neurons leading to a reduced number of dendrites [113]. *Shi* is Drosophila's dynamin, a GTPase necessary for endocytosis and vesicle

recycling [114, 115]. It regulates endocytosis throughout its development [116]. Temperature-sensitive *shi* mutants *shits1*, *shits3* and *shits6* have been reported to display phenotypes of embryonic lethality, continuous larval, and adult paralysis at 29˚C [117, 118]. Similarly, loss of function of *shi* results in disruption of the tracheal network with ectopic branching and misal-location of dorsal trunk cells, implicating *shi* in tracheal development [119].

In summary, HEAL was able to predict important genes involved in development or immune response conditions, which were not previously identified in Drosophila melanogaster. This discovery will provide more insight into the immune response factors and the growth mechanism in Drosophila. Furthermore, the success of the HEAL model has provided a viable solution to the challenge of limited class labelled data to train a ML classifier often encountered in bioinformatics predictive analysis.

## Conclusion

We developed a heuristic-enabled active machine learning model that eliminates the human component in the active learning pipeline and possessing a superior prediction performance compared to the state-of-the-art AL models based on five public datasets. The HEAL model was implemented as a web tool for annotating biological and non-biological data when there is limited labelled data for training a machine learning model. The HEAL model was also applied to address the problem of predicting conditionally essential genes which is an intractable problem in bioinformatics. Essential immune response and embryonic developmental stage genes in *D. melanogaster* were predicted. Four of the 10 predicted immune response genes were novel and 53 genes were identified as important in the embryonic developmental stage in *D. melanogaster*. These predicted genes are proposed for future experimental studies.

## Supporting information

**S1 Fig. Confusion matrix from the comparative analysis of HEAL and other techniques on five real-world datasets.** A. HEAL results show a significantly lower false-positive rate except on the cancer dataset. B. UncAL produces higher true positives except on the Cancer and Credit-A data. C. RandAL performed well on the Credit-G data with a higher true positive and lower false negative.
(DOCX)

**S1 Table. The complete list of predicted genes that play essential roles during the embryonic stage of *D. melanogaster*.**
(CSV)

## Author Contributions

**Conceptualization:** Olufemi Tony Aromolaran, Marcus Oswald, Ezekiel Adebiyi, Rainer Koenig, Jelili Oyelade.

**Data curation:** Olufemi Tony Aromolaran, Eunice Adedeji, Rainer Koenig.

**Formal analysis:** Olufemi Tony Aromolaran, Itunu Isewon, Marcus Oswald, Rainer Koenig, Jelili Oyelade.

**Funding acquisition:** Ezekiel Adebiyi, Rainer Koenig.

**Investigation:** Jelili Oyelade.

**Methodology:** Olufemi Tony Aromolaran, Marcus Oswald, Ezekiel Adebiyi, Jelili Oyelade.

**Project administration:** Itunu Isewon, Ezekiel Adebiyi, Rainer Koenig, Jelili Oyelade.

**Resources:** Ezekiel Adebiyi, Rainer Koenig.

**Software:** Olufemi Tony Aromolaran.

**Supervision:** Rainer Koenig, Jelili Oyelade.

**Validation:** Olufemi Tony Aromolaran, Eunice Adedeji, Jelili Oyelade.

**Visualization:** Olufemi Tony Aromolaran, Jelili Oyelade.

**Writing – original draft:** Olufemi Tony Aromolaran, Itunu Isewon, Eunice Adedeji, Marcus Oswald, Ezekiel Adebiyi, Rainer Koenig, Jelili Oyelade.

**Writing – review & editing:** Olufemi Tony Aromolaran, Itunu Isewon, Eunice Adedeji, Marcus Oswald, Ezekiel Adebiyi, Rainer Koenig, Jelili Oyelade.

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
