## [Decision Letter · Decision Letter 0]

25 Apr 2023

PONE-D-23-10047Heuristic-enabled active machine learning for conditionally essential gene prediction in D. melanogasterPLOS ONE

Dear Dr. Oyelade,

Thank you for submitting your manuscript to PLOS ONE. After careful consideration, we feel that it has merit but does not fully meet PLOS ONE’s publication criteria as it currently stands. Therefore, we invite you to submit a revised version of the manuscript that addresses the points raised during the review process.

We look forward to receiving your revised manuscript.

Kind regards,

Jian Xu, Ph.D.

Academic Editor

PLOS ONE

Journal Requirements:

https://pubmed.ncbi.nlm.nih.gov/34471501/

https://bmcbioinformatics.biomedcentral.com/articles/10.1186/s12859-017-1745-2

https://ieeexplore.ieee.org/document/7081610

In your revision ensure you cite all your sources (including your own works), and quote or rephrase any duplicated text outside the methods section. Further consideration is dependent on these concerns being addressed.

   "This work was supported by the Deutsche Forschungsgemeinschaft (https://www.dfg.de/) within the project KO 3678/5-1, and the German Federal Ministry of Education and Research (BMBF, Fkz 01EO1002, 01EO1502 and 13N15711) and funding from the World Bank awarded to Covenant Applied Informatics and Communication Africa Centre of Excellence (CApIC-ACE) through the ACE Impact Project (2019 – 2024)."

   "1. Deutsche Forschungsgemeinschaft (https://www.dfg.de/) within the project KO 3678/5-1, and the German Federal Ministry of Education and Research (BMBF, Fkz 01EO1002, 01EO1502 and 13N15711)

2. World Bank awarded to Covenant Applied Informatics and Communication Africa Centre of Excellence (CApIC-ACE) through the ACE Impact Project (2019 – 2024)

Reviewers' comments:

Reviewer's Responses to Questions

**Comments to the Author**

1. Is the manuscript technically sound, and do the data support the conclusions?

Reviewer #1: Partly

2. Has the statistical analysis been performed appropriately and rigorously? 

Reviewer #1: No

3. Have the authors made all data underlying the findings in their manuscript fully available?

Reviewer #1: No

4. Is the manuscript presented in an intelligible fashion and written in standard English?

Reviewer #1: Yes

5. Review Comments to the Author

Reviewer #1: In the article “Heuristic-enabled active machine learning for conditionally essential gene prediction in D. melanogaster”, Aromolaran and colleagues propose a machine learning approach supported by heuristics to support the annotation and employ the prediction of conditional essential genes. The authors attempt to eliminate the curation of gene essentiality by using an uncertainty-based learning approach. While there is merit in tackling a challenging topic, there are methodological concerns considering the manuscript and data/code availability as they are.

-The concepts regarding conditional essentiality were well defined, but there is important literature on the topic that should be reviewed and or discussed:

https://www.sciencedirect.com/science/article/abs/pii/S0959437X18301291

https://www.sciencedirect.com/science/article/pii/S0734975021001282

https://pubmed.ncbi.nlm.nih.gov/35939967/

https://pubs.rsc.org/en/content/articlelanding/2009/mb/b905264j

-Curation of potentially essential/non-essential and conditional essential genes in D. melanogaster based on FlyBase annotations has been performed here and should be discussed:

https://academic.oup.com/nargab/article/2/3/lqaa051/5874956

-It is important to highlight that defining conditional essentiality experimentally is very challenging, requiring functional testing under a range of conditions and accounting for potential bias in the functional techniques employed as well as standard annotations.

-The authors should be more clear that they are seeking to automate data labeling (and the curation of genes for essentiality in the end) while still using an active learning approach. Sometimes they suggest they are replacing traditional ML approaches, which is not true or the point is not clear (e.g. 74-79).

- For clarity, the Materials and methods should appear before the Results (suggestion)

-The authors evaluate the performance of their proposed method using and comparing other selection approaches and data available in the literature. However, the data and source code for the HEAL and for the ML approach are not available at the moment. Only the pseudocode for the HEAL is available. The repository only contains binary files and code related to the website, and the repository is not well documented. ALL the source code (ML/bash commands/software versions) and aLL data used for the manuscript should be public and clearly documented, for scientific reproducibility and scrutiny. Use Github for the code and Figshare/Zenodo for the data (if too big for Github).

-The authors should focus the paper on the proposed HEAL approach and validate with solid benchmark datasets, potentially involving gene essentiality data (not only the 5 presented, or justify why these 5 datasets are enough to validate their approach). The authors should present the conditional essentiality application as a use-case, not as the main topic, particularly because it is difficult to validate the results. The structure of the manuscript is confusing as is.

-The authors manually define sets of essential/non-essential and conditional genes 377-386. However, the terms used to select/label those genes within FlyBase are not clear. These annotations are critical for the success of the approach and should be well documented. What defines essentiality for embryonic development and/or immune response? This should be supported by scientific literature. How can the authors be sure that they are selecting conditional essential genes? Or the authors are focussing on genes that are essential for specific conditions (embryonic development and immune response)? This is quite confusing and should be clarified and consistent in the manuscript.

-Why using essentiality for immune response? The focus seems to be essentiality for survival(?). This is not clear and not defined anywhere. If the focus is on essentiality for immune response, this should be raised and discussed in the introduction.

-Features were extracted for the genes but the code used for this has not been made available.

-Cross-validation results using the selected gene essentiality data should be demonstrated using a figure. The data/code for the cross-validation should be available as well. The validations using the gene essentiality data are weak in the current form.

-Only 11 of the 151 predicted to be essential for embryonic development have support from FlyBase. How can the authors be confident about the others? Again, the validation here appears weak.

- GO analysis for genes essential for embryonic development should be included in the title (line 577-578)

-The authors should register a domain name for http://165.73.223.20. This is because any IP address change would disrupt the availability of the website.

-It is clear that some well-defined method for curation of data is still necessary prior to employing HEAL, and this should be acknowledged by the authors (Discussion/Conclusion).

-The authors discuss some genes identified by their approach at top-ranked. Were these genes included in the training set or in the test set only?

-The authors should clearly discuss the limitations of the present study

-If validated, the HEAL method can be used for binary classification approaches beyond conditional essentiality. This can be raised in the Discussion

-More information on the HEAL website implementation should be given (Python/Django?)

6. PLOS authors have the option to publish the peer review history of their article (what does this mean?). If published, this will include your full peer review and any attached files.

Reviewer #1: No

---

## [Author Response · Author response to Decision Letter 0]

6 Jun 2023

Reviewer #1: In the article “Heuristic-enabled active machine learning for conditionally essential gene prediction in D. melanogaster”, Aromolaran and colleagues propose a machine learning approach supported by heuristics to support the annotation and employ the prediction of conditional essential genes. The authors attempt to eliminate the curation of gene essentiality by using an uncertainty-based learning approach. While there is merit in tackling a challenging topic, there are methodological concerns considering the manuscript and data/code availability as they are.

-The concepts regarding conditional essentiality were well defined, but there is important literature on the topic that should be reviewed and or discussed:

https://www.sciencedirect.com/science/article/abs/pii/S0959437X18301291

https://www.sciencedirect.com/science/article/pii/S0734975021001282

https://pubmed.ncbi.nlm.nih.gov/35939967/

https://pubs.rsc.org/en/content/articlelanding/2009/mb/b905264j

-Curation of potentially essential/non-essential and conditional essential genes in D. melanogaster based on FlyBase annotations has been performed here and should be discussed:

https://academic.oup.com/nargab/article/2/3/lqaa051/5874956

Response: We thank the reviewer for pointing us to the related literature. We have reviewed these publications in line 23 -41 and 66-74

-It is important to highlight that defining conditional essentiality experimentally is very challenging, requiring functional testing under a range of conditions and accounting for potential bias in the functional techniques employed as well as standard annotations.

Response: We appreciate the reviewer for highlighting this fact. We have included this fact in line 72-74.

-The authors should be more clear that they are seeking to automate data labeling (and the curation of genes for essentiality in the end) while still using an active learning approach. Sometimes they suggest they are replacing traditional ML approaches, which is not true or the point is not clear (e.g. 74-79).

Response: To clarify the statement in line 351-353, the paragraph stated the goal of the study to replace the human component in the Active learning approach and not replace ML approaches.

- For clarity, the Materials and methods should appear before the Results (suggestion)

Response: The sections have been arranged as suggested by the reviewer.

-The authors evaluate the performance of their proposed method using and comparing other selection approaches and data available in the literature. However, the data and source code for the HEAL and for the ML approach are not available at the moment. Only the pseudocode for the HEAL is available. The repository only contains binary files and code related to the website, and the repository is not well documented. ALL the source code (ML/bash commands/software versions) and aLL data used for the manuscript should be public and clearly documented, for scientific reproducibility and scrutiny. Use Github for the code and Figshare/Zenodo for the data (if too big for Github).

Response: We appreciate the author for pointing out the packaging of our data and source code, we now provide all the data and source with clear description in github.

-The authors should focus the paper on the proposed HEAL approach and validate with solid benchmark datasets, potentially involving gene essentiality data (not only the 5 presented, or justify why these 5 datasets are enough to validate their approach). The authors should present the conditional essentiality application as a use-case, not as the main topic, particularly because it is difficult to validate the results. The structure of the manuscript is confusing as is.

Response: We agree with the reviewer as regards the need to present the conditional essentiality as a use case and this is evident in the title modification. We chose not to validate with other gene essentiality data because the focus of the study is modifying active learning for labeling biological when there is limited training data which also helps to focus the paper on the proposed HEAL approach as suggested.

-The authors manually define sets of essential/non-essential and conditional genes 377-386. However, the terms used to select/label those genes within FlyBase are not clear. These annotations are critical for the success of the approach and should be well documented. What defines essentiality for embryonic development and/or immune response? This should be supported by scientific literature. How can the authors be sure that they are selecting conditional essential genes? Or the authors are focussing on genes that are essential for specific conditions (embryonic development and immune response)? This is quite confusing and should be clarified and consistent in the manuscript.

Response: Again, we thank the reviewer for pointing this out. This study is focusing on predicting conditionally essential genes given 2 specific conditions. The data collected from Flybase relates to lethality during the 2 conditions. More description about the flybase annotation for the two conditions in focus has been added in line 142-147.

-Why using essentiality for immune response? The focus seems to be essentiality for survival(?). This is not clear and not defined anywhere. If the focus is on essentiality for immune response, this should be raised and discussed in the introduction.

Response: As stated above, the essentiality relates to immune response condition. This is now discussed in the introduction section on line 43-49.

-Features were extracted for the genes but the code used for this has not been made available.

Response: Publicly available tools were used for feature generation and these tools were described in the Feature generation section line 156 of the manuscript.

-Cross-validation results using the selected gene essentiality data should be demonstrated using a figure. The data/code for the cross-validation should be available as well. The validations using the gene essentiality data are weak in the current form.

Response: We acknowledge the concerns of the reviewer. We would like to point out that Fig. 1 is a figure that graphically presents the cross-validation results as requested by the reviewer however, the same cannot be graphically presented for the Drosophila data because we only know the label for about 10% of the data unlike ML approach where the label for both training and test data is available prior to the analysis. Attempting to use ML performance metrics like precision-recall or accuracy will give a false representation of the model performance. To ensure the HEAL model performance is comparable to existing models, we used the UCI data for validation of the model. 

-Only 11 of the 151 predicted to be essential for embryonic development have support from FlyBase. How can the authors be confident about the others? Again, the validation here appears weak.

Response: We thank the reviewer for pointing us in this direction. After careful observation of the result, we discovered that predictions with probability score lower than 0.6 were erroneously reported. We did describe the reason for using the decision boundary of 0.6 in line 559-561. We now excluded genes with probability score lower than 0.6 thereby reducing the number of predictions from 151 to 53 with 5 genes validated in Flybase (Table S1). We thereafter search the literature to find the link between the top 10 predicted genes and developmental stage condition. Our findings are found in the discussion section. A potential limitation of the HEAL model is also mentioned in the discussion section.

- GO analysis for genes essential for embryonic development should be included in the title (line 577-578)

Response: The Title has been modified to reflect the scope of the study

-The authors should register a domain name for http://165.73.223.20. This is because any IP address change would disrupt the availability of the website.

Response: We thank the reviewer for this suggestion, the domain name is now registered.

-It is clear that some well-defined method for curation of data is still necessary prior to employing HEAL, and this should be acknowledged by the authors (Discussion/Conclusion).

Response: The methods for data curation were presented with relevant references in the methods section

-The authors discuss some genes identified by their approach at top-ranked. Were these genes included in the training set or in the test set only?

Response: The identified genes were not included in the training set. 

-The authors should clearly discuss the limitations of the present study

Response: We thank the reviewer for this comment. A paragraph about the limitation has been included in the discussion section line 485-494

-If validated, the HEAL method can be used for binary classification approaches beyond conditional essentiality. This can be raised in the Discussion

-More information on the HEAL website implementation should be given (Python/Django?)

Response: We have now included more information on the website implementation on line 414-418.

---

## [Decision Letter · Decision Letter 1]

19 Jun 2023

Heuristic-enabled active machine learning: A case study of predicting essential developmental stage and immune response genes in Drosophila melanogaster

PONE-D-23-10047R1

Dear Dr. Oyelade,

We’re pleased to inform you that your manuscript has been judged scientifically suitable for publication and will be formally accepted for publication once it meets all outstanding technical requirements.

Kind regards,

Jian Xu, Ph.D.

Academic Editor

PLOS ONE

Additional Editor Comments (optional):

Reviewers' comments:

Reviewer's Responses to Questions

**Comments to the Author**

1. If the authors have adequately addressed your comments raised in a previous round of review and you feel that this manuscript is now acceptable for publication, you may indicate that here to bypass the “Comments to the Author” section, enter your conflict of interest statement in the “Confidential to Editor” section, and submit your "Accept" recommendation.

Reviewer #1: All comments have been addressed

2. Is the manuscript technically sound, and do the data support the conclusions?

Reviewer #1: Yes

3. Has the statistical analysis been performed appropriately and rigorously? 

Reviewer #1: Yes

4. Have the authors made all data underlying the findings in their manuscript fully available?

Reviewer #1: Yes

5. Is the manuscript presented in an intelligible fashion and written in standard English?

Reviewer #1: Yes

6. Review Comments to the Author

Reviewer #1: The authors have addressed all comments and concerns carefully. I consider that the manuscript is now suitable for publication.

7. PLOS authors have the option to publish the peer review history of their article (what does this mean?). If published, this will include your full peer review and any attached files.

Reviewer #1: No

---

## [Editor Report · Acceptance letter]

31 Jul 2023

PONE-D-23-10047R1 

Heuristic-enabled active machine learning: A case study of predicting essential developmental stage and immune response genes in *Drosophila melanogaster*

Dear Dr. Oyelade:

I'm pleased to inform you that your manuscript has been deemed suitable for publication in PLOS ONE. Congratulations! Your manuscript is now with our production department. 

Kind regards, 

on behalf of

Dr. Jian Xu 

Academic Editor

PLOS ONE